# CALIBRATION OF NEURAL NETWORK LOGIT VECTORS TO COMBAT ADVERSARIAL ATTACKS.

## ABSTRACT

Adversarial examples remain an issue for contemporary neural networks. This paper draws on Background Check (Perello-Nieto et al., 2016), a technique in model calibration, to assist two-class neural networks in detecting adversarial examples, using the one dimensional difference between logit values as the underlying measure. This method interestingly tends to achieve the highest average recall on image sets that are generated with large perturbation vectors, which is unlike the existing literature on adversarial attacks (Cubuk et al., 2017). The proposed method does not need knowledge of the attack parameters or methods at training time, unlike a great deal of the literature that uses deep learning based methods to detect adversarial examples, such as Metzen et al. (2017), imbuing the proposed method with additional flexibility.

## 1    INTRODUCTION AND RELATED WORK

Adversarial examples are specially crafted input instances generated by adversarial attacks. The term was introduced by Szegedy et al. (2013) in the context of image classification. These attacks generate, or manipulate data, to achieve poor performance when classified by neural networks, which poses existential questions about their usage in high stakes security critical applications. Since they were introduced, there have been many papers that have introduced novel attack methods and other papers that attempt to combat those attacks. For instance, Goodfellow et al. (2014) introduces the fast gradient sign method (FGSM), and Papernot et al. (2016c) proposes a method based on modifying the gradient of the softmax function as a defence.

Adversarial attacks can be identified into various classes such as white box and black box, where in the former, the attack has full knowledge of all model parameters. Examples created by these attacks can be false positives or false negatives. In the case of images, they can be nonsensical data (e.g. noise classified as a road sign) or clear cut (e.g. a visually clear cat, classified as a road sign). These attacks can be non-targeted or targeted such that the classifier chooses a specific class for the adversarial example. Various adversarial defences exist, some based on deep learning techniques and others on purely distributional techniques. Similar work on adversarial defences has been done by Grosse et al. (2017), in which the network is trained on specific attack types and parameters with an additional outlier class for adversarial examples. A multi-dimensional statistical test over the maximum mean discrepancy and the energy distance on *input* features is then used to classify instances as adversarial. Other work has been done by Bradshaw et al. (2017), where Gaussian Processes are placed on top of conventional convolutional neural network architectures, with radial basis kernels, imbuing the neural network with a way of understanding its own perceptual limits. The authors find that the network becomes more resistant to adversarial attack. The work that follows continues in a similar vein to both of these methods. Some methods such as Metzen et al. (2017) use sub-units of deep learning architectures to detect adversarial instances.

Calibration is a technique of converting model scores, normally, through application of a post processing function, to probability estimates. Background Check is a method to yield probability estimates, via a set of explicit assumptions, in regions of space where no data has been observed. In this work, Background Check is useful in producing calibrated probabilities for adversarial data that often exists in regions where no training and test data has been seen. Reliable probability estimates can then be measured by calibration and refinement loss. Various calibrating procedures exist such as binning, logistic regression, isotonic regression and softmax. Jordan et al. (1995) demonstrates

the logistic function is optimal when the class-conditional densities are Gaussians with unit variance. Softmax extends this to multi-variate Gaussian densities with unit variance.

Calibration of neural network models has been performed by Guo et al. (2017), using a method called Temperature Scaling, that modifies the gradient of the softmax function allowing softmax to calibrate densities with non-unit variance. The authors perform this calibration after noticing that calibration loss for neural networks has increased in recent years. When adversarial attacks against neural networks are brought into perspective, a problem arises for existing calibration techniques, which is the question of mapping adversarial logit scores to reliable probability estimates (which should be zero for a successful adversarial attack). In this work, a method is demonstrated that uses Background Check to identify adversarial attacks.

## 2 CALIBRATION

A classifier is said to be well calibrated, if, as the number of predictions approaches infinity, the proportion of outcomes given probability $p$, occur $p$ fraction of the time. Denoting $x_1, x_2, ..., x_n$ as data-set instances and $y_1, y_2, ..., y_n$ as their corresponding ground truth class labels, a scoring classifier $s = f(x_i)$ has a calibrating function $\mu$ applied to it yielding $\mu(f(x_i))$. Perfect calibration is defined as the expectation $s_i = f(x_i)$ such that $s_i = \mathbb{E}[Y|f(X) = s_i]$ where random variables $X, Y$ denote the features and class label of a uniformly randomly drawn instance from the data-set respectively, such that $Y = 1$, $Y = 0$ represent an individual positive and negative instance, respectively.

To visualize calibration performance of a classifier, Murphy & Winkler (1977) plots the observed frequency of an event against the predicted frequency yielding calibration curves. Calibration curves plot the observed relative frequency against the predicted probability for all test data. Perfect calibration occurs if the calibration curve exactly fits the identity line. Refinement loss measures the difference between a probability estimate and zero or one.

This can be combined with a frequency distribution, giving an indication of spread, which is useful if there are few events associated with a particular probability. The notion of refinement is also useful when considering calibration. By considering the crude constant classifier, which predicts the probability corresponding to the class distribution for all inputs, it is clear that this calibration estimate is perfectly calibrated. However, an intuitively more valuable calibration estimate, is one which predicts a value closer to either zero or one. For this reason, DeGroot & Fienberg (1981) suggests measuring refinement loss, which measures the distance of the classifiers probability estimates to either zero or one. Together, calibration and refinement loss make up the Brier Score. Kull & Flach (2015) defines calibration and refinement loss.

*Calibration loss* = $\mathbb{E}[d(S, C)]$ is the loss due to the expected difference between the model score $S$ and the proportion of positives among instances (observed relative frequency) with the same score.

*Refinement loss* = $\mathbb{E}[d(C, Y)]$ is the loss due to the presence of instances from multiple classes among instances with the same estimate $S$. In the worst case, this clearly reduces to the crude constant classifier mentioned above.

An instance of recent work related to calibration is Beta calibration. Beta calibration Kull et al. (2017) is based on the beta distribution which includes functions such as the logit, sigmoid and identity. This allows it to calibrate scores produced by models such as naive bayes, which biases its scores towards extremities when the assumption of feature independence is not met, using the inverse sigmoid or logit function, a function which is not in softmax's repertoire.

## 3 ADVERSARIAL ATTACKS

In the context of adversarial attacks, the optimization problem that is often formulated to construct adversarial examples is shown.

$$\arg\min_{\delta_x} ||\delta_x|| \, s.t. F(X + \delta_x) = Y^*$$

An adversarial example $X^*$ is constructed by adding a perturbation vector $\delta_x$, where $Y^*$ is the desired adversarial output and $X + \delta_x \in [0, 1]$ where $[0, 1]$ is the upper and lower bound of a well

formed example, subject to the constraints above. Intuitively, these constraints coerce the network into mis-classifying each example with a minimal perturbation vector. Example distance metrics that measure the size of the perturbation include $\mathcal{L}_p$ metrics, as well as the PASS score Rozsa et al. (2016), a metric designed to better reflect notions of psycho-physical similarity than $\mathcal{L}_p$ metrics.

Adversarial attack methods include L-BFGS from Szegedy et al. (2013), which uses linear trial and error to find a $c$ for each data instance such that $c$ multiplied by an arbitrary small perturbation vector $\epsilon$ mis-classifies the instance. FGSM builds on the L-BFGS method, replacing expensive linear search with gradient descent to find the perturbation vector. This uses the following optimization strategy to create the perturbation vector $\eta$, such that $x$ is the input to the model, $y$ is the predicted class, $\theta$ are the model parameters and $J(\theta, x, y)$ is the cost function. $\triangledown_x$ indicates the derivative of the cost function is taken with respect to the input $x$. The sign function takes the sign of the resulting derivative. The adversarial example is then constructed as $x' = x + \eta$.

$$\eta = \epsilon sign(\triangledown_x, J(\theta, x, y))$$

A momentum term can be added to the gradient descent process to yield the work of Dong et al. (2018). The BIM attack by Kurakin et al. (2016), uses a smaller noise vector $\epsilon$ produced by FGSM, applied iteratively, before a clip operation is applied to the resulting image, keeping it within the maximum image pixel values after each iteration.

JSMA is a forward derivative approach, by Papernot et al. (2016b), which uses a Jacobian matrix, to produce an adversarial saliency map, which indicate the features that when (positively or negatively) perturbed, most efficiently achieve a desired network output. DeepFool (Moosavi Dezfooli et al., 2016) finds an image that is at a minimal distance to the decision boundary from the proposed example, to another target class which isn't the source class, treating multi-class classifiers as combinations of binary affine classifiers.

## 4 BACKGROUND CHECK

Background Check, introduced by Perello-Nieto et al. (2016), is the calibrating procedure that our proposed method uses to defend neural networks from adversarial attacks. Background Check takes as input, the scores from logit vectors and maps them to probabilities, replacing softmax *after training*. Background Check provides a framework to classify regions where no previous data has been seen as background regions, that in the context of the proposed method will be classified as regions where adversarial data may lie.

Background Check also provides a framework to resolve ambiguity inherent in a single value representing probability. More specifically, Background Check provides two values to represent a single probability value of a data instance. One value represents distance from the data density, and another represents the certainty of a particular class. This approach avoids overloading the meaning of a single number representing probability. More specifically, an uncertainty of $1/n$ for all classes, could represent an instance very close to the decision boundary or very far away from training data. On the other hand, an output of zero, could represent a point very far away from training data or a classification that the data is definitely not that particular class.

### 4.1 BACKGROUND CHECK IN MODELLING SPARSE DATA

Background Check introduces an additional outlier background class, $b$, representing regions of space where data is sparse or non-existent. Then, a foreground class is introduced, $f$. This represents regions of space where data is plentiful or dense, i.e. data from any class apart from the background class, is abundant. $b$ is introduced as an additional class whilst $f$ is kept as a reference class. Every instance $x$ necessarily belongs to either $f$ or $b$. $P(b|x) = 0$ and $P(f|x) = 1$ refers to absolute certainty that the instance belongs to one of the classes with *sufficient* training data, where $P(C|x)$ is a conditional probability measure. The ratio of the two conditional measures defines the reliability factor $r(x)$.

$$r(x) = \frac{P(f|x)}{P(b|x)} = \frac{P(f_1|x) + ... + P(f_n|x)}{P(b|x)} = \frac{P(f,x)P(x)}{P(b,x)P(x)} = \frac{P(x,f)}{P(x,b)}$$

If $r(x) \leq 1$, the classification that the reliability factor indicates is $b$, else if $r(x) > 1$ then the classification is $f$. $P(x, f)$ and $P(x, b)$ are referred to as the foreground and background densities. Furthermore, the relative foreground and background densities can be defined, $q_f(x)$ and $q_b(x)$.

$$q_f(x) = \frac{P(x, f)}{\max_x P(x, f)}, \qquad q_b(x) = \frac{P(x, b)}{\max_x P(x, f)}$$

Intuitively, the relative density outputs the proportion of f or b, at the point in space corresponding to the instance $x$ being evaluated. Simple dividing $q_f(x)$ by $q_b(x)$ yields $r(x)$.

### 4.1.1   JUSTIFICATION OF $q_b(x)$

To construct $q_b(x)$, four inductive biases, in increasing strength, are given. This work only uses the third inductive bias.

1.  *Inductive bias 1* : $q_b(x)$ is a function of $q_f(x)$. This is justified, by the idea that with no other information, there is no reason to assign different background densities to points with the same foreground density. The domain knowledge informs the function used $\mu$ : $[0, 1] \longrightarrow [0, \infty)$.
2.  *Inductive bias 2* : monotonicity of $\mu$, that is, when moving to a region with higher foreground density the background increases or decreases.
3.  *Inductive bias 3* : an affine bias, i.e. $\mu(x) = ax + b$ or by replacing a and b: $\mu(x) = (1 - x)\mu(0) + x\mu(1)$.
4.  *Inductive bias 4* : constant background ie $\mu(0), \mu(1) = 0.5$.

### 4.1.2   IMPLEMENTATION OF BACKGROUND CHECK

The key notion in Background Check is the implementation of the inductive bias that shapes $q_b$. This implementation is provided in two different different ways.

1.  *BCD* : referred to as the discriminative approach, involves generating artificial background instances around foreground data and then training a binary discriminative classifier to separate them. The instances are generated in a hypercube or a hypersphere, such that the background is half as dense as $max_x P(x, f)$.
2.  *BCF* : referred to as the familiarity approach, this involves fitting a one class model on the foreground data to obtain $q_f$, then using an inductive bias to obtain $q_b$. The data, $x$, that is being fit must have an underlying measure. The familiarity factor $r(x)$ can be found, allowing, the posterior probabilities $P(b|x)$ and $P(f|x)$ to be computed.

The implementation that the proposed method in this work uses, is the BCF method for its speed in high dimensional spaces. The measure underlying the space was the one-dimensional $L_1$ difference between elements of the logit vector. For instance, given a two-class score vector $[-5, 5]$, then the score difference is 10. This measure represents the distance of a data point to the softmax decision boundary. The one-class model used to fit $q_f(x)$ was a gamma function optimized using maximum likelihood estimation. To establish the link from $q_f(x)$ to $q_b(x)$, the third inductive bias was used with $\mu(0) = 1$ and $\mu(1) = 0$, with domain knowledge informing the use of a power value. This link manifests itself in the equation below.

$$q_b = (1 - q_f(x))^5 \times \mu(0) + q_f(x) \times \mu(1)$$

## 5   EXPERIMENTS

One neural network for each attack type, parameter combination and dataset, is trained with the Adam optimizer (Kingma & Ba (2014)). A batch size of 256 and a learning rate of 0.001 is employed. The biases of the neurons are set to 0 and the weights are sampled uniformly at random

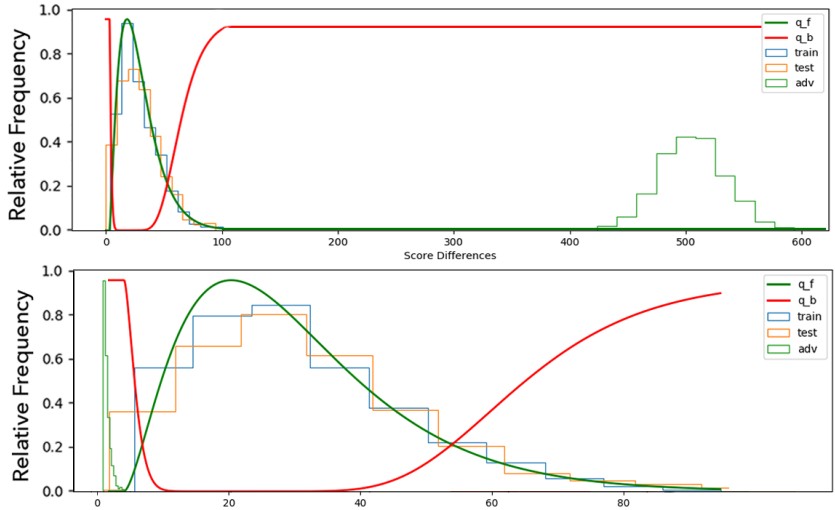

Figure 1: Background Check applied to the score difference of two, two class neural network logit vectors (one network each for the top and bottom figures). The training, test and adversarial images, in blue, orange and green, respectively, are organised into ten bins each. The adversarial data for the top figure was generated by the momentum attack, with very large perturbations applied. The bottom figure has adversarial data crafted by the JSMA attack with a moderate perturbation vector applied. The green line represents $q_f$, the foreground relative density and $q_b$ represents the background relative density. It is clear, that the adversarial images in both cases are distinctly separated from the training and test data, which overlap to the point of in-distinguishability. In the top figure, the adversarial images have logit differences much larger than the training/test data compared to much smaller logit differences for the bottom figure.

from the interval $[0, 1]$. Each neural network is tasked with classifying a variety of paired class combinations from the CIFAR10 dataset. These pairs were: dog versus plane (DVP), fish vs ship (FVS) and airplane vs horse (AVH). Regularization techniques such as $L_2$ regularization and dropout, with $p = 0.5$ were applied to the networks. The training vs test ratio is 5:1, such that each class has exactly 1000 test images and 5000 training images. The images were pre-processed such that all image pixel values were floating point values between zero and one, rather than values between 0 and 255. Seven different adversarial attacks, from the cleverhans API (Papernot et al., 2016a), were tested against the networks with a variety of parameter combinations. The adversarial images were generated from test data. The attacks were all white-box attacks and performed on the network which included a final softmax layer in its structure. The final two-class average recall of each network on the validation set of the network was always above 80% after only 300 iterations over the training data. The attacks had different effects on each image, for identical parameter settings and as such, the parameters were subjectively chosen based on whether the image fell into one of four classes. Some attacks, due to a mixture of constraints and difficulty in searching the adversarial image space, only had images in one or two of the available classes.

1. *Large* - Image consists entirely of noise.

2. *Typical* - Moderate noise levels, underlying image recognizable.

3. *Small* - Recognizable noise, but clear image.

4. *Very Small* - No noticeable noise, clear image.

For instance, BIM an iterative method had 10 iterations, with $\epsilon : 0.8$ and $\epsilon_i : 0.05$, for a large perturbation vector, yet the method from Miyato et al. (2015), a non-iterative method had $\epsilon : 12.0$. The full parameter settings are listed in the appendix.

The average recall, defined in the equation below, rather than accuracy, is evaluated due to the presence of varying class proportions. The 2-class average recall is on the two classes in CIFAR-10.

Table 1: Network architecture used for each of the networks.

| Layer | Units | Layer Type | Kernel | Activation | Stride |
|-------|-------|-----------|--------|-----------|--------|
| 1 | 32 | Convolutional | 5x5 | ReLu | 1 |
| 2 | n/a | Pooling | 2x2 | Max | 2 |
| 3 | 64 | Convolutional | 5x5 | ReLu | 1 |
| 4 | n/a | Pooling | 2x2 | Max | 2 |
| 5 | 256 | Fully Connected | n/a | ReLu | n/a |
| 6 | 2 | Fully Connected | n/a | Linear | n/a |

Table 2: Table of results

| Method / Perturbation size | 2-class average recall | **3-class average recall with Background Check** | Difference | **Adversarial TPR** | CIFAR-10 Pair |
|---|---|---|---|---|---|
| BIM / Large | 91% | 81% | -10 | 100% | DVP |
| BIM / Typical | 93% | 88% | -5 | 100% | DVP |
| BIM / Small | 90% | 66% | -24 | 56% | DVP |
| BIM / Very Small | 91% | 58% | -33 | 21% | DVP |
| Mom. / Large | 87% | 85% | -2 | 100% | AVH |
| Mom. / Typical | 82% | 79% | -3 | 93% | FVS |
| Mom. / Small | 88% | 71% | -17 | 63% | DVP |
| Mom. / Very Small | 91% | 55% | -36 | 7% | DVP |
| Madry / Large | 95% | 89% | -6 | 100% | DVP |
| Madry / Typical | 89% | 85% | -4 | 100% | DVP |
| Madry / Small | 88% | 60% | -28 | 28% | DVP |
| Madry / Very Small | 91% | 53% | -38 | 23% | DVP |
| FGSM / Large | 94% | 77% | -17 | 61% | DVP |
| FGSM / Typical | 87% | **87%** | 0 | 99% | FVS |
| FGSM / Small | 95% | 58% | -37 | 16% | DVP |
| VAT / Large | 94% | 71% | -23 | 47% | DVP |
| VAT / Typical | 93% | 66% | -27 | 34% | DVP |
| DeepFool / Typical | 91% | 84% | -7 | 98% | DVP |
| JSMA / Typical | 85% | **87%** | +2 | 97% | FVS |
| Baseline | 50% | 33% | -17 | 33% | DVP |
| No Adversarials | 94% | 88% | -6 | 100% | DVP |

The 3-class average recall includes the adversarial class.

$$\left(\sum_i^C \frac{TP_i}{TP_i + FN_i}\right) \times {}^1\!/_C$$

## 6 RESULTS

The table of results demonstrates that large perturbation vectors, associate with a mean reduction in average recall of 11.6, whereas for very small perturbation vectors, the mean reduction in average recall is 35.7. Typical perturbation vectors have a mean reduction in average recall of 6.9. In all cases except for two, the average recall decreases. It is clear that the adversarial TPR generally increases as the size of the perturbation vector increases. In particular, three out of the five large perturbation vectors achieve a TPR on the adversarial class of 100%. All models achieve higher average recall than the baseline, which was simply a strategy that with uniform randomness guesses the class.

In order to visualize areas where Background Check assigns foreground and background densities, it is helpful to construct histograms. When constructing histograms of the test, training and adversarial $L_1$ logit differences, three categories were established over the space.

1. Adversarial examples in a distinct cluster, closer to the decision boundary, than the training and test data.

2. Adversarial examples scattered amongst the training/test data.

3. Adversarial examples in a distinct cluster, further from the decision boundary, than the training and test data.

The JSMA and DeepFool attacks found logit differences smaller than the test and training logits, yet still high enough to yield a significant confidence level when applied to the softmax function. The Madry, Momentum and BIM attacks produced logit differences far higher than the test and training logit differences. However, some attacks found logit differences within the test and training distributions.

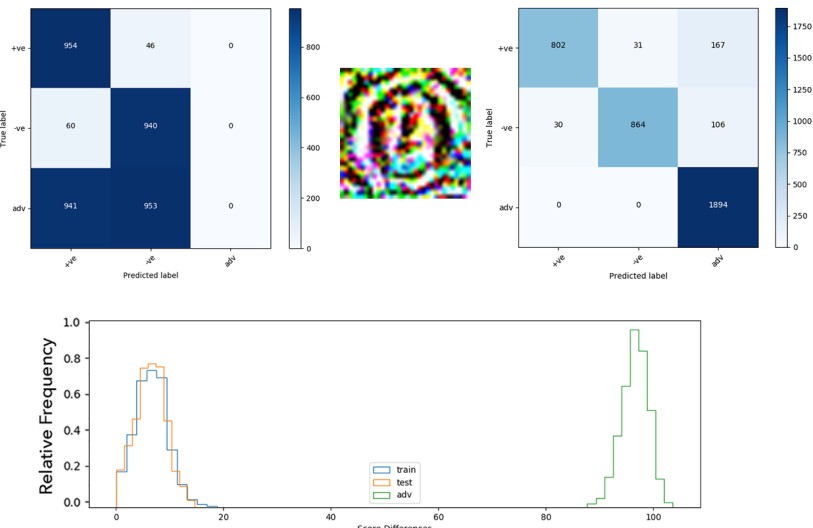

Figure 2: Madry attack from Madry et al. (2017) in category 3, with a large perturbation vector applied. The differences between the per-class logit values are large, at least 70, albeit less than some of the other attacks which have score differences in the hundreds. The adversarial images with scores represented by the green histogram are very far away from the training and test data, which means that Background Check will separate them well. The central example successful adversarial image is noise, due to the large perturbation applied. The top left confusion matrix is that of the neural network without Background Check. The top right confusion matrix is that of the neural network with Background Check applied to it. The confusion matrices are relatively coloured with the true labels on the vertical axis and the predicted labels on the horizontal axis. The histogram in the bottom of each image uses *ten* bins, whose size is chosen by scipy, relative to the spread of the data. The y-axis or height of each bar of the blue, orange and green histograms represent the relative frequency of examples in the training, test and adversarial classes, respectively. The x-axis represents the score difference between the logit vectors. The left hand confusion matrix, on the bottom row, shows the attack had equivalent ease generating adversarial images from either class. The right hand confusion matrix shows a reduction in average recall due to the many, 198/1000 false negatives, for the first class, and 136/1000 for the second, which, because the classifier is evaluated on the average recall, will make a large difference to the final average recall. For the histograms of all of the methods, please see the appendix.

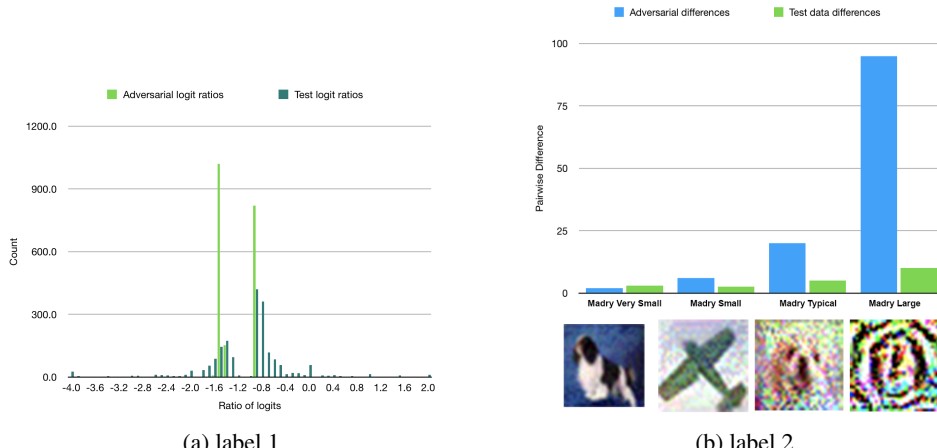

(a) label 1             (b) label 2

Figure 3: Figure a) shows that the ratio of positive class values versus negative class values for adversarial data and test data. Adversarial ratios are concentrated around 1 and -1, whereas the test logit ratios assume values from -4 to + 2. Figure b) shows the trend of the pairwise differences as the attack parameters increase for the attack from Madry et al. (2017)

# 7 DISCUSSION

Background check models the data density over the logit differences. It is clear from the figures in the appendix that the adversarial attacks find examples in regions where the test and training data do not exist. Thus, Background Check improves the discriminative performance of the classifier when dealing with adversarial examples with large perturbation vectors. These resulting images are noisy and hard to allocate a non-ambiguous class, though we argue that these images can occur and be just as damaging in the real world and as such need to be defended against. It would be useful to follow on from this method with an analysis of generative probability estimation and a corresponding measure of calibration and refinement loss.

Promising future research would scale up the logit difference metric underlying Background Check to higher dimensional spaces to deal with a full ten classes to allow for comparability to mainstream literature on adversarial defences. Possible metrics that can underlie Background Check could use the energy distance. In addition, Background Check could be applied to each layer of a neural network. However, this must be setup such that it does not interfere with the ability of the neural network to generalize.

The performance of Background Check can be measured in different ways. For example, Condessa et al. (2017) proposes performance measures for classification systems with the rejection option. These measures consist of metrics such as the *non-rejected accuracy*, which measures the ability of the classifier to accurately classify non-rejected samples. The *classification quality*, which measures the correct decision making of the classifier with the rejector and finally, *the rejection quality*, which measures the ability to concentrate all mis-classified samples onto the set of rejected samples.

# 8 CONCLUSION

A novel approach to defending neural networks against adversarial attacks has been established. This approach intersects two previously unrelated fields of machine learning, calibration and adversarial defences, using the principles underlying Background Check. This work demonstrates that adversarial attacks, produced as a result of large perturbations of various forms, can be detected and assigned to an adversarial class. The larger the perturbation, the easier it was for the attacks to be detected.

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

Table 3: Table of results

| Attack Type / Perturbation Size | Parameter Combination |
|---|---|
| BI / Large | (nb : 10, $\epsilon$ : 0.8, $\epsilon_i$ 0.05) |
| BI / Typical | (nb : 10, $\epsilon$ : 0.3, $\epsilon_i$ 0.05) |
| BI / Small | (nb : 10, $\epsilon$ : 0.1, $\epsilon_i$ 0.05) |
| BI / V Small | (nb : 10, $\epsilon$ : 0.06, $\epsilon_i$ 0.05) |
| DeepFool, JSMA / Typical | Default |
| Madry / Large | (nb : 150, $\epsilon$ : 8.0, $\epsilon_i$ 0.03) |
| Madry / Typical | (nb : 40, $\epsilon$ : 0.3, $\epsilon_i$ 0.01) |
| Madry / Small | (nb : 40, $\epsilon$ : 0.1, $\epsilon_i$ 0.01) |
| Madry / V Small | (nb : 10, $\epsilon$ : 0.05, $\epsilon_i$ 0.01) |
| VAT / Large | $\epsilon$ : 12.0 |
| VAT / Typical | $\epsilon$ : 1.0 |
| FGSM / Large | $\epsilon$ : 200.0 |
| FGSM / Typical | $\epsilon$ : 0.3 |
| FGSM / Small | $\epsilon$ : 0.05 |
| Momentum / Large | (nb : 10, $\epsilon$ : 16.0, $\epsilon_i$ 2.0) |
| Momentum / Typical | (nb : 10, $\epsilon$ : 0.3, $\epsilon_i$ 0.06) |
| Momentum / Small | (nb : 10, $\epsilon$ : 0.15, $\epsilon_i$ 0.06) |
| Momentum / V Small | (nb : 10, $\epsilon$ : 0.1, $\epsilon_i$ 0.03) |

# Appendices

## A   PARAMETERS

$\epsilon$ is the size of the perturbation, $\epsilon_i$ is the change of the size each iteration and nb is the number of iterations.

## B   EXTRA DISTRIBUTIONS

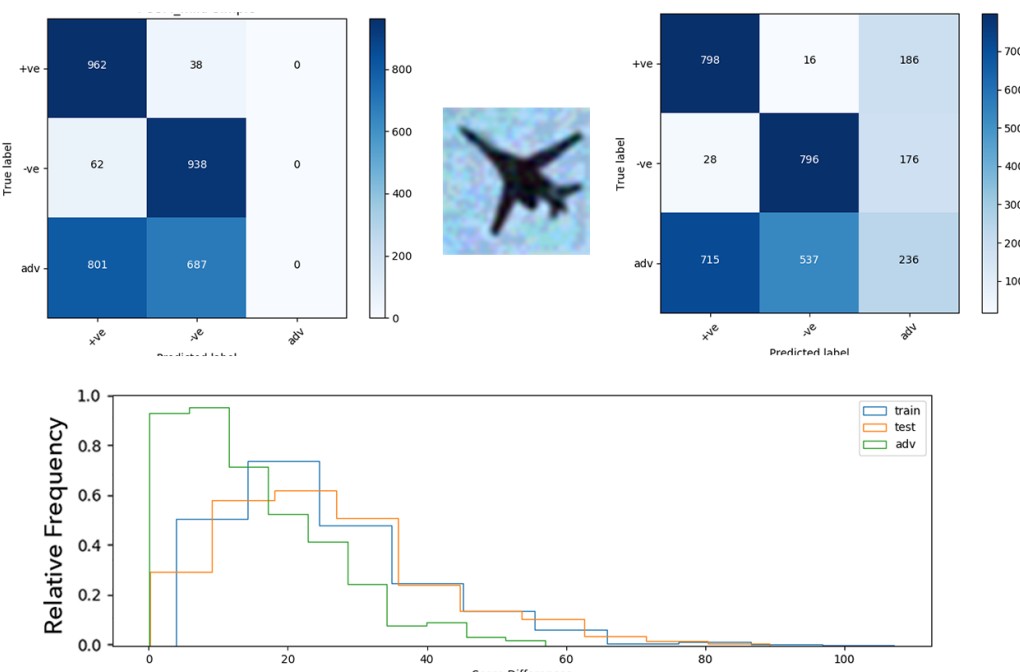

Figure 4: Attack from Goodfellow et al. (2014) in category 2, otherwise known as the FGSM attack, with smaller than typical parameters. The analysis is similar to that in Figure 2 except for a few important differences. The first, is that the adversarial examples represented by the green histogram, clearly overlap with the training and test distributions in this space. This makes it very hard for Background Check to separate these examples. This pattern occurred mainly for attacks with smaller than usual parameters. This is exemplified by the right hand confusion matrix classifying only 16% of the adversarials correctly. For more analysis, see the caption in Fig. 2.

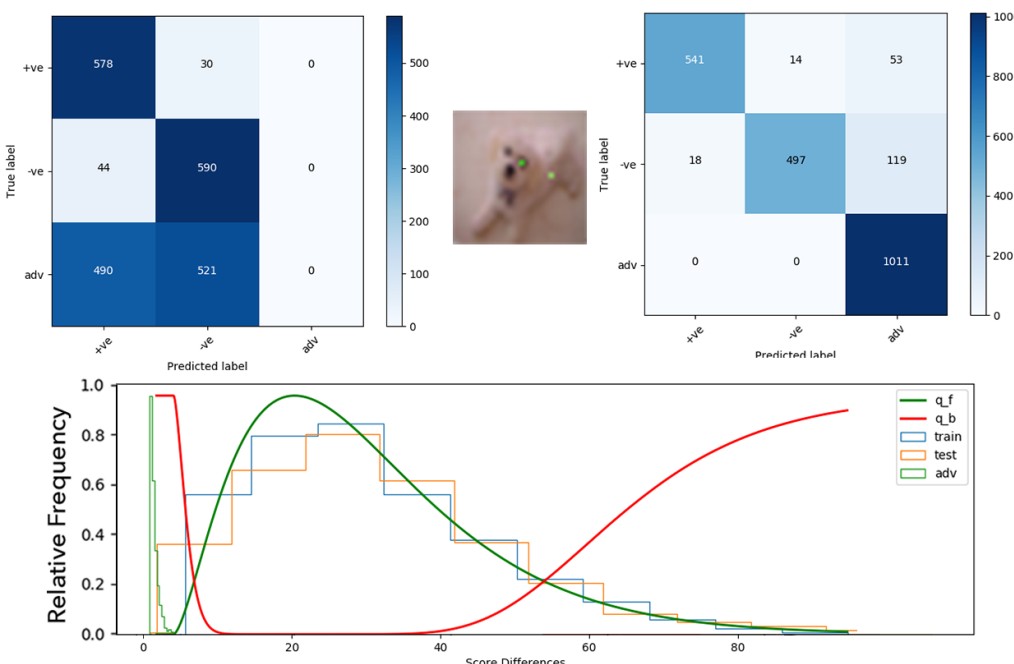

Figure 5: Attack from Papernot et al. (2016b), otherwise known as the JSMA attack, in category 1, with typical parameter settings, with $q_b$ and $q_f$ shown. The adversarial differences here are smaller than training and test differences, allowing Background Check to separate them well. JSMA only changes two pixels in this attack.

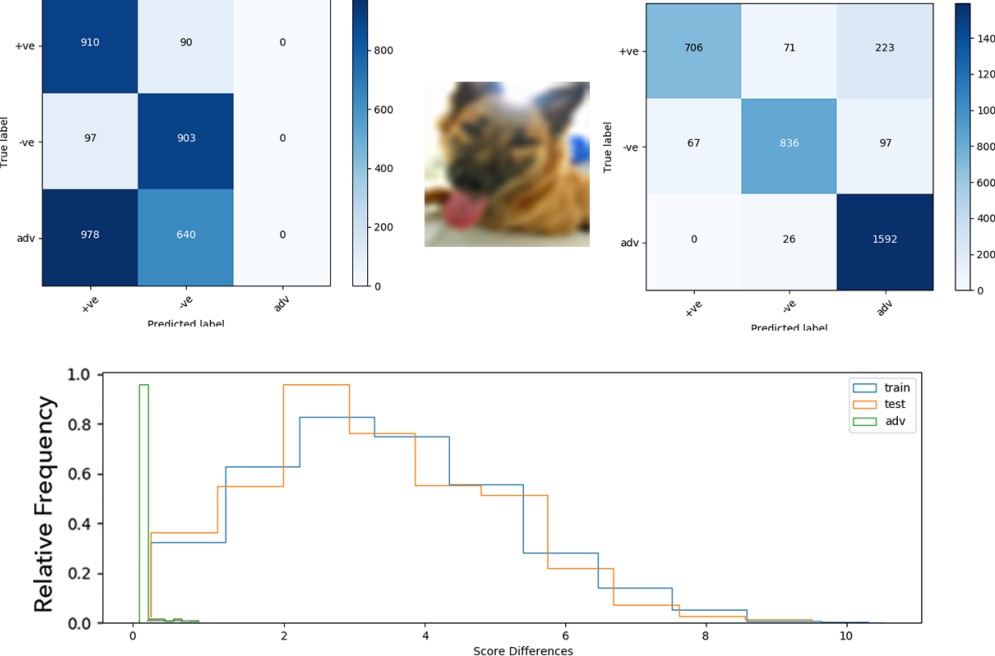

Figure 6: Attack from Moosavi Dezfooli et al. (2016) (DeepFool) in category 1, with typical parameters. The adversarial differences here are also smaller than the training and test differences, allowing Background Check to separate them well. DeepFool finds the minimum perturbation vector to misclassify an image, hence, it appears the images are close to the softmax decision boundary. For more analysis, see the caption in Fig. 2.

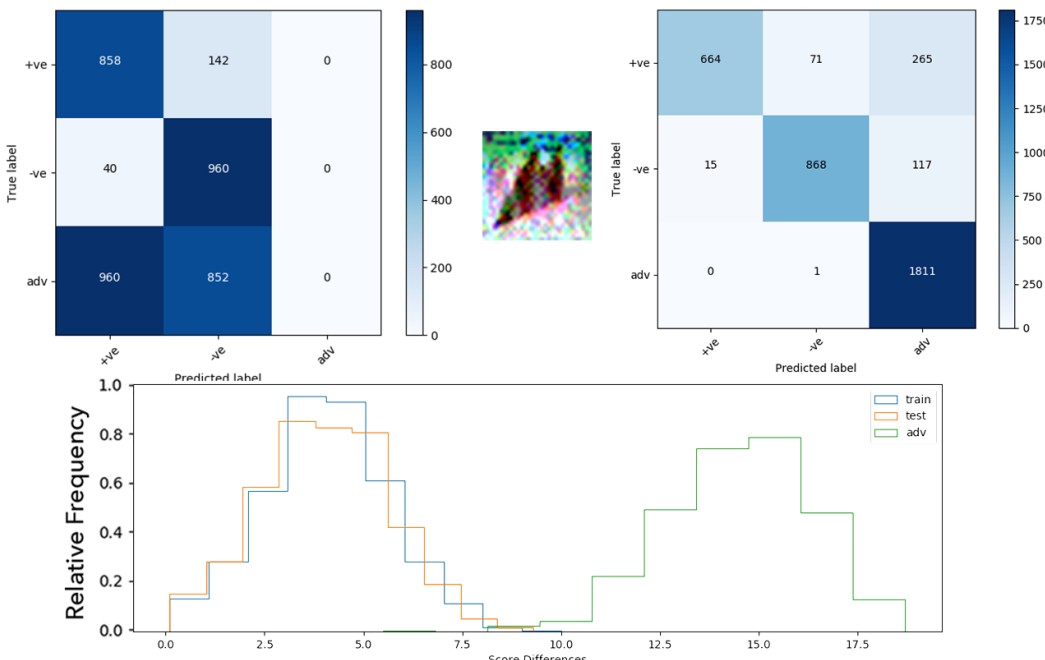

Figure 7: Attack from Dong et al. (2018), otherwise known as the momentum attack, in category 3, with typical parameter settings. Here, the clusters are separated well, leading to an effective separation by Background Check. For more analysis, see the caption in Fig. 2.

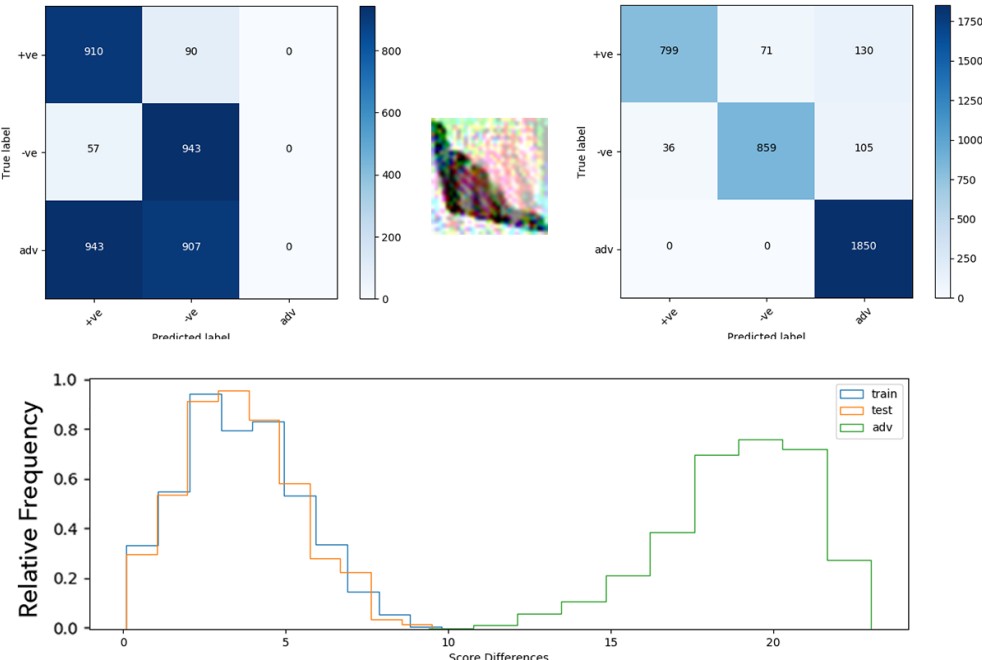

Figure 8: Attack from Kurakin et al. (2016), otherwise known as the Basic Iterative (BI) attack. in category 3, with typical perturbation. Once again, here, the clusters are separated well, leading to an effective separation by Background Check. For more analysis, see the caption in Fig. 2.

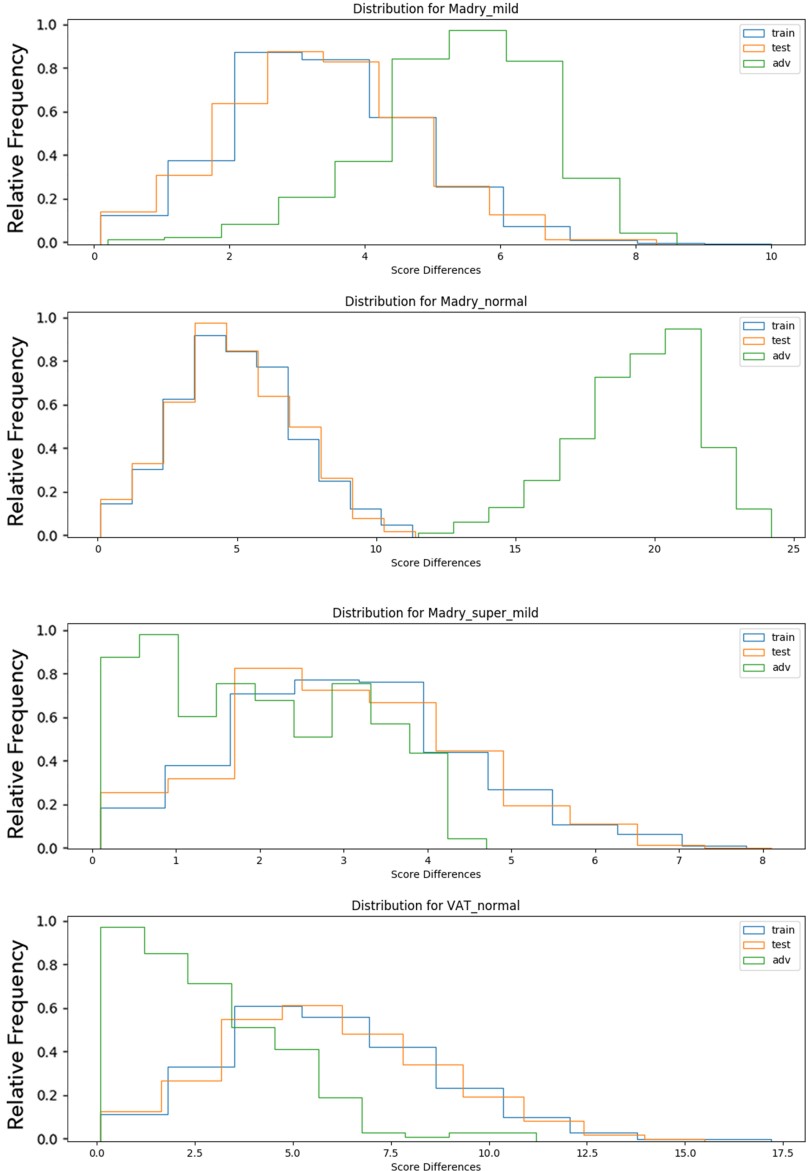

Figure 9: Extra distribution figures. These are the logit difference distributions figures for all attacks not in figures in the results section. They show the histograms of where the images sit in logit space, for adversarial sets in green, training sets in blue and test sets in orange. The figures show varying degrees of overlap in the histograms, which often correlate with the efficacy of the adversarial defence proposed in this work.

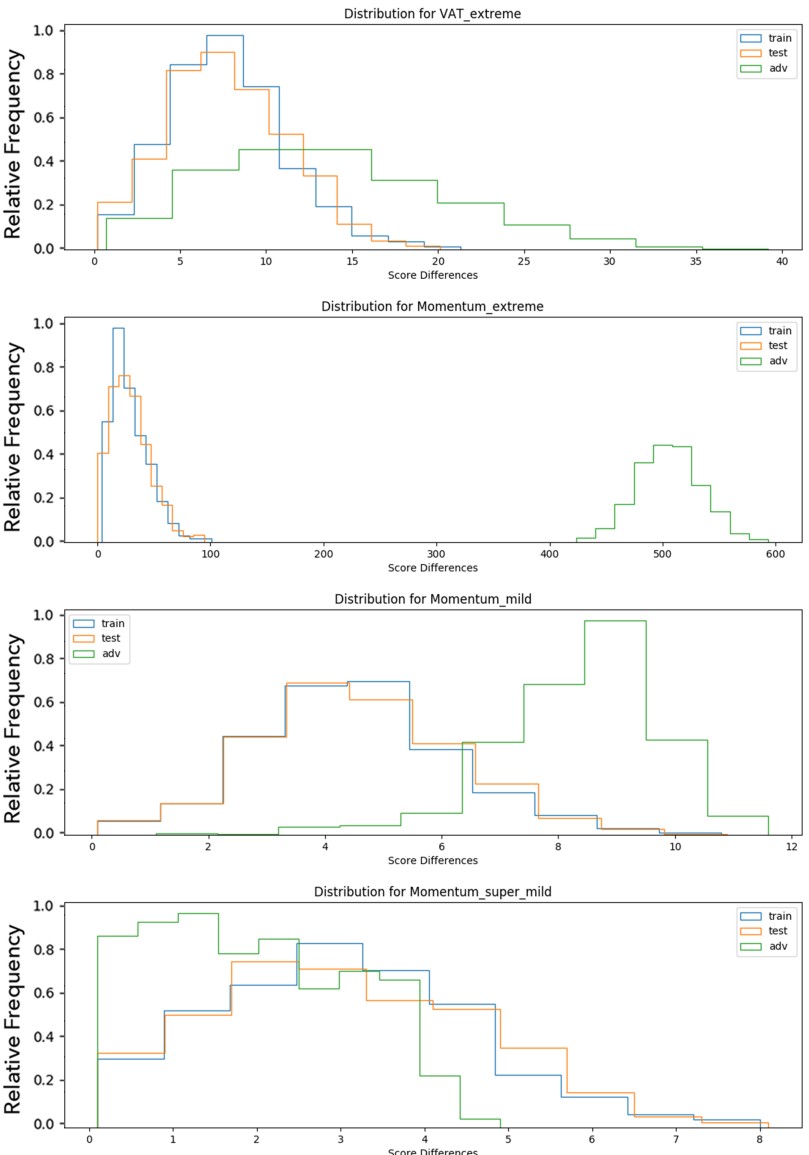

Figure 10: Extra distribution figures

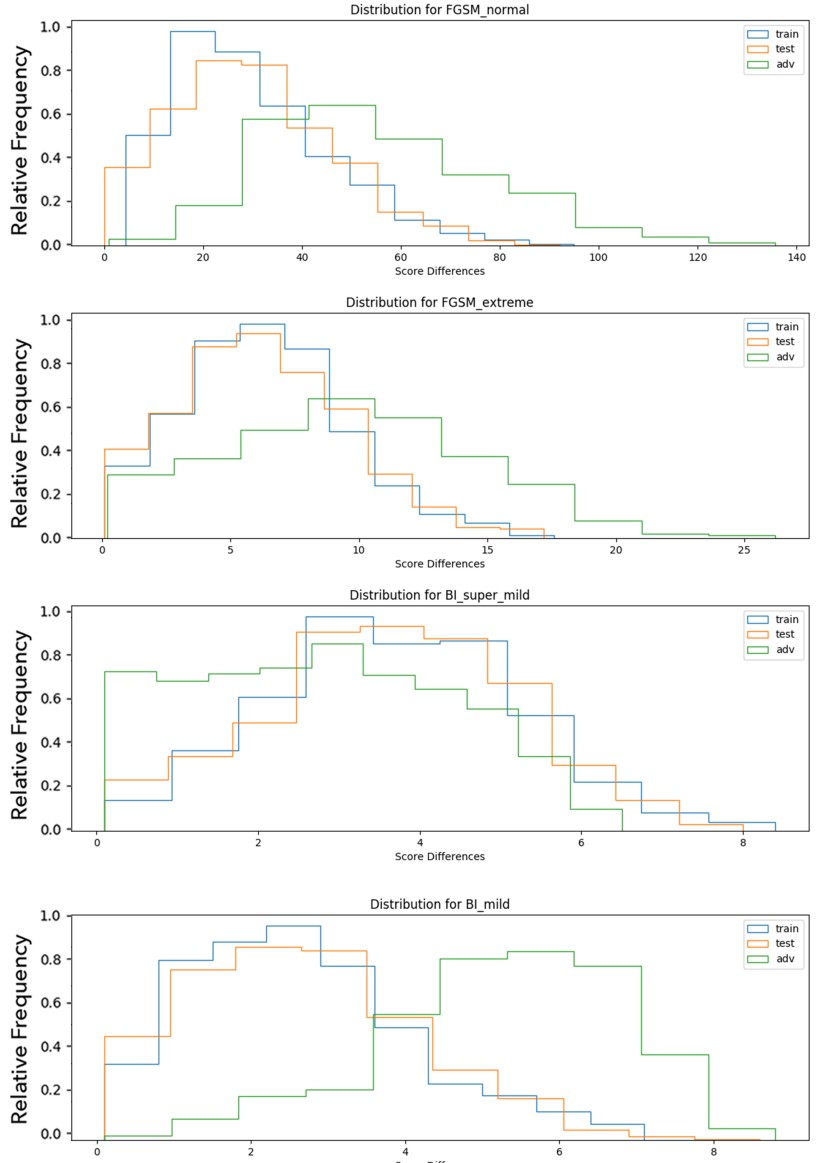

Figure 11: Extra distribution figures

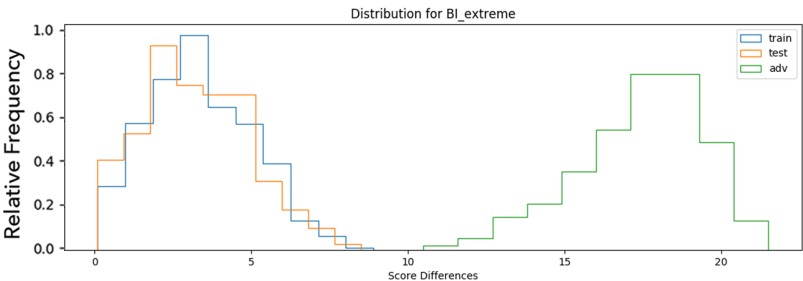

Figure 12: Extra distribution figures

