# OpenReview forum: "Calibration of neural network logit vectors to combat adversarial attacks"
_ICLR.cc/2019/Conference_

### Official Review · AnonReviewer3 · 2018-10-31

**Rating:** 4
**Confidence:** 5

**Review:**

Strength:

Intuition that multiple sources of uncertainty are relevant to adversarial examples

Weaknesses:

Threat model is unclear
No adaptive adversaries are considered
Attack parameters could be better justified

The intuition presented at the beginning of Section 4 is interesting. There are indeed multiple sources of uncertainty in machine learning, and the softmax probability only captures confidence partially. In particular, estimating the support of training data for a particular prediction and the density of that support is conceptually relevant to understanding and mitigating outlier test points like adversarial examples.

Given that the approach is motivated as a defense (see Section 7 for instance), it needs to be evaluated in a realistic adversarial setting. In particular, it would greatly strengthen the paper if a clear threat model was specified. In your rebuttal, would you be able to formulate clearly what adversarial capabilities and goals were assumed when designing this defense?

All experiments are performed on a binary variant of CIFAR-10. In addition, all pairs chosen for the experiments are well-separated: dogs are semantically further apart from airplanes than they are from horses. Would you be able to clarify in your rebuttal how the approach would generalize to multi-class classification?

Perhaps the strongest limitation of the evaluation is that it does not consider adaptive adversaries. This goes back to the threat model point raised previously. Adaptive strategies will be put forward by adversaries aware of the defense being deployed (security should not be obtained through obscurity). For instance, the adversary could modify their attack to have it minimize the difference between logits on the training and adversarial data. This would help evading detection by the proposed scheme. However, results from Section 6 are shown for attacks that do not attempt to reduce the L1 difference between adversarial and training data.

Some attack parameters could also be better justified. The naming convention for the perturbation sizes reads a bit imprecise and is perhaps more confusing than it is informative. Furthermore, could you explain in your rebuttal why epsilon is larger than 1.0 for the FGSM---when the inputs where normalized between 0 and 1?

Details:

Page 1: Typo in “defence”
Page 2: Notation s_i is overloaded multiple times making it difficult to parse expressions
Page 2: Citation to Kull et al. should use \citep after “Beta calibration”
Page 3: Citation to Rozsa et al. should use \citep after “PASS score”
Page 5: Generally-speaking, it’s best to compute attacks at the logit layer rather than the probabilities to avoid numerical instabilities, which can then lead to gradient masking. However, the following sentence suggests the opposite: “The attacks were all white-box attacks and performed on the network which included a final softmax layer in its structure.”

---

> ### Author Response · Authors · 2018-11-18
> **Response**
>
> Thanks you for your comments. It is a simple technique and I appreciate the highlighted shortcomings mentioned. I agree with all of the points mentioned.

---

### Official Review · AnonReviewer1 · 2018-10-31
**Lack of clarity and weak experimental evaluation**

**Rating:** 2
**Confidence:** 4

**Review:**

On the positive side, I think it's a good idea to experiment with various approaches to defend DNNs against adversarial attacks, like the Background Check approach considered in this manuscript (which hasn't gotten a lot of traction in the Machine Learning community so far).

However, the manuscript has a number of shortcomings which in my opinion makes it a strong rejection. My main concern is about the experimental evaluation:
- The authors should test their approach on Carlini & Wagner's attack which allows for explicit control of logit differences and thus could entirely defeat the Background Check.
- Moreover, any paper on this topic should evaluate defenses in a complete white-box setting, i.e. the adversary is aware of the detection method and actively tries to bypass it.
- A comparison with other detection methods from the literature is missing, too, and the two-class classifier setting is very limited.

Besides that, I find there is a general lack of clarity:
- It really becomes clear only towards the end of the paper what the Background Check is applied for, namely, the detection of adversarial samples. This should be clearly articulated from the beginning.
- Notation isn't always properly introduced (e.g. in the formula for 3-class average recall on page 6), and the same goes for
some acronyms (e.g. what is TPR?).
- Where does Table 2 show a "mean reduction in average recall of 11.6", and what does that mean exactly?

---

> ### Author Response · Authors · 2018-11-18
> **Response**
>
> Thanks you for your comments. It is a simple technique and I appreciate the highlighted shortcomings mentioned. I agree with all of the points mentioned.

---

### Official Review · AnonReviewer2 · 2018-11-05
**Density estimation for adversarial regions seems beneficial, but a stronger justification is needed**

**Rating:** 3
**Confidence:** 4

**Review:**

This paper addresses adversarial detection through the absolute-value difference between the two logit vector values of a DNN binary classifier, with one class associated with normal data and the other with adversarial data. Assignment of examples to an "adversarial" class is problematic in that adversarial examples are typically generated in regions for which training data is very sparse. To cope with this, the authors propose use of the Background Check calibration techniques recently proposed by Perello-Nieto et al. (ICDM 2016).

Here, BC is used to estimate probabilities in a sparse "background" class (here, the adversarial class)  through a form of interpolation based on foreground and background densities. The underlying distributional assumption used for estimating foreground densities was that of a gamma function.  Rather than using BC's affine bias for estimating background density from the foreground density, the authors adapt it by raise the weighting for the "adversarial" decision to the fifth power. Unfortunately, no justification for this choice is given, other than to say that this was done with "domain knowledge informing the use of a power value".

In their experimentation, the authors generate from CIFAR-10 data four kinds of adversarial attacks: noise alone, images with moderate noise, clear images with noticeable noise, and clear images with imperceptible noise. For a variety of attacks, they showed (in Table 2) differences between the average recall for normal examples vs the average recall for normal plus adversarial images. However, without knowing the proportion of adversarial examples used in testing, the significance of the reported differences cannot be judged. They also list the true positive rates of adversarial examples, which showed much variation from experiment to experiment (trending to rather poor performance for attacks with imperceptible noise). Again, the significance of the results cannot be judged without knowing the false negative rate, true negative rate, etc. Moreover, the results are reported without clearly identifying two of the attacks used ("Mom." is presumably Dong et al.'s attack using momentum in gradient descent, and Miyato et al.'s "VAT" is not properly introduced in the related work). Crucially, no evaluation of their method is made with respect to other adversarial detection strategies.

Pros:
* Overall, the calibration approach is well motivated, and likely to be of some benefit.
* The paper is generally readable and understandable. The issues behind calibration and the use of BC are well explained.

Cons:
* The result is a simple and straightforward application of an existing technique - not greatly original.
* Many design choices in the model (particularly the raising of one of the weights to a seemingly-arbitrary power) are mysterious. No indication is given as to other alternatives or how they might perform.
* The experimental results are inadequate to judge the impact of the proposed calibration approach.
* There is no comparison against other detection methods.

---

> ### Author Response · Authors · 2018-11-18
> **Response**
>
> Thanks you for your comments. It is a simple technique and I appreciate the highlighted shortcomings mentioned. I agree with all of the points mentioned.

---

### Public Comment · (anonymous) · 2018-10-24
**No adaptive attack?**

Much prior work has already shown FGSM/PGD/JSMA can be detected when the adversary is not attempting to evade the defense (Metzen et al. 2017, Grosse et al. 2017). However, Carlini & Wagner 2017 showed that almost all of these detection schemes could be easily bypassed if the attacker actively attempts to evade the defense.

Have you tried such an attack?

Nicholas Carlini and David A. Wagner. Adversarial examples are not easily detected: Bypassing ten detection methods. AISec 2017.

---

> ### Author Response · Authors · 2018-10-27
> **Confirmation that no adaptive attack was performed against the defense.**
>
> This threat model was not considered in the context of this defense. An attack that optimised for this defense, would attempt to find examples with similar logit differences to the test and training data. This could very well happen. This paper demonstrates and then exploits the fact that the logit differences for adversarial examples tend to be far larger or far smaller than the training and test data. Your criticism of the nature of the empirical result in this paper is valid. Future work would definitely explore such a threat model.

---

### Meta-Review · Area_Chair1 · 2018-12-12
**Limited contribution and no effort for rebuttal**

**Confidence:** 5
**Recommendation:** Reject

**Metareview:**

The reviewers and AC note the potential weaknesses of the paper in various aspects, and decided that the authors need more works to publish.